# Automating multi-label crisis detection in psychological support hotlines with pre-trained models

**Shuying Rao**[1,2,3], **Guifeng Deng**[1,2,3], **Haidong Song**[1], **Qiuxia Chen**[1], **Mingjin Luo**[1], **Yaoyun Zhang**[4], **Sha Zhao**[3,5], **Gang Pan**[3,5], **Tao Li**[1,3,6]*, **Haiteng Jiang**[1,3,6]*

1 Affiliated Mental Health Center and Hangzhou Seventh People's Hospital and School of Brain Science and Brain Medicine, Zhejiang University School of Medicine, Hangzhou, China, 2 College of Biomedical Engineering and Instrument Science, Zhejiang University, Hangzhou, China, 3 Liangzhu Laboratory, MOE Frontier Science Center for Brain Science and Brain-machine Integration, State Key Laboratory of Brain-machine Intelligence, Zhejiang University, Hangzhou, China, 4 School of Biomedical Informatics, University of Texas at Dallas, Richardson, Texas, United States of America, 5 College of Computer Science and Technology, Zhejiang University, Hangzhou, China, 6 Zhejiang Key Laboratory of Clinical and Basic Research for Psychiatric Diseases, Hangzhou, China

* h.jiang@zju.edu.cn (HJ); litaozjusc@zju.edu.cn (TL)

## Abstract

Psychological support hotlines provide immediate help to individuals in crisis, with operators assessing emotional states and suicide risk. However, increasing demand has led to a shortage of trained professionals, emphasizing the need for AI-driven crisis detection models. This study included 1,057 calls from the Hangzhou Hotline (2022–2023) to evaluate the effectiveness of deep learning and pre-trained models in detecting psychological crises using audio (Wave2Vec, Whisper) and transcribed text (RoBERTa, GPT). We adopted two strategies: deep learning classification with pre-trained models and Large Language Models (LLMs)-based prediction via prompt engineering (GPT-4 and DeepSeek series). The deep learning framework, employing GPT embeddings excelled in multi-label predictions compared to auditory model, achieving 80.48% [80.18%, 80.78%] F1 scores for identifying high-risk calls in prospective tests. Fusion experiments revealed that acoustic features offered negligible predictive value compared with text semantics. Notably, GPT-4o and DeepSeek-R1, utilizing few-shot learning, demonstrated performance comparable to the GPT-embedding deep learning model across multiple tasks. This suggests that their advanced Chain-of-Thought reasoning effectively mitigates data dependency gap, enabling LLMs to align with clinical domains using a few examples. Expert evaluation confirmed the clinical applicability of GPT-generated explanations. Taken together, these findings highlight the potential of LLMs in mental health crisis detection and lay the foundation for future research.

**Data availability statement:** The datasets analyzed during the current study are not publicly available due to the sensitive nature of the data, which involves privacy and confidentiality protections for psychological support hotline users. Data are available upon reasonable request for researchers who meet the criteria for access to confidential data. The data are securely stored on the institutional servers of Hangzhou Seventh People's Hospital. Data access requests may be sent to the Ethics Committee of Hangzhou Seventh People's Hospital at hqyllbgs@163.com.

**Funding:** This work was supported by the National Science and Technology Major Project (2022ZD0212400 to HJ; 2021ZD0200404 to TL), the National Natural Science Foundation of China (82371453 to HJ), and the Key Research and Development Program of Zhejiang Province (2024C03006 to HJ; 2026C01013 to TL; 2024ZY01010 to TL; 2024E10107 to TL). The funders had no role in study design, data collection and analysis, decision to publish, or preparation of the manuscript.

**Competing interests:** The authors have declared that no competing interests exist.

## Author summary

Psychological support hotlines provide essential immediate care for individuals in distress; however, the rising volume of calls frequently overwhelms the limited number of trained professionals. In this study, we explored how deep learning models utilizing pre-trained embeddings and large language models (LLMs) can support these services by automatically detecting severe emotional crises and suicide risks. By analyzing over 1,000 real hotline calls collected over two years at a large hospital in Hangzhou, we compared deep learning models using pre-trained audio and text embeddings against LLMs utilizing prompt engineering, aiming to simultaneously flag emotional crisis and potential self-harm risk. We found GPT text embeddings-based deep learning frameworks consistently outperformed audio-based models, achieving a higher average accuracy and F1 score in depressed emotion detection and risk assessment. Also, LLMs such as GPT-4o and DeepSeek-R1 matched the performance of embedding-based frameworks while learning from very few examples. Crucially, human experts confirmed that the explanations provided by the LLMs was clinically relevant. These findings indicate that LLMs could act as a vital "co-pilot" for hotline operators, helping to prioritize urgent calls, though further research is needed to address clinical safety and reliability prior to real-world deployment.

## Introduction

Suicide and suicidal behaviors present formidable challenges for public health and policy. According to the World Health Organization, suicide claimed over 720,000 lives globally [1], while in China, more than 100,000 suicides and approximately 1 million hospital visits for suicide attempts are reported annually [2]. Particularly concerning is the rising suicide rate among adolescents [3], which highlights an urgent need for effective prevention strategies. Depression and other psychological crisis factors are key contributors to suicide, leading to complex psychological struggles and depletion of energy, which in severe cases can lead to the tragic end of death [4]. The WHO's "Live Life" framework highlights early identification of individuals at risk of suicide is a cornerstone of successful intervention [5]. This underscores the critical importance of timely and accurate identification of depression and suicidal behaviors as a foundation for prevention efforts.

Psychological support hotlines have been established in many countries, ranging from developed to developing nations, as a crucial component of suicide prevention strategies [6,7]. These services connect individuals in crisis with trained professionals, offering immediate counseling, comprehensive risk assessments, and necessary interventions [8,9]. However, the growing demand for mental health services has outpaced available resources [10], leading to a shortage of trained personnel and inconsistent care due to the lack of standardized risk assessment tools [11]. These challenges underscore the pressing need for innovative solutions to enhance hotline

operations. Developing and validating an automated Intelligence tool for suicide risk assessment could address these issues by improving the efficiency of resource allocation, reducing response times, and ensuring equitable access to interventions for at-risk individuals.

Hotline data provides a valuable resource for suicide risk assessment and prevention [12], as it inherently reflects the mental distress of callers who voluntarily seek help. These calls are digitally recorded, offering extensive insights into mental health dynamics across a wide spectrum of topics. Furthermore, callers often express negative emotions that can signal depression or heightened suicide risk [13,14]. Previous studies have demonstrated that individuals exhibiting suicidal ideation often show distinct vocal patterns, such as reduced speech rate and lower volume [15]. Detecting and analyzing these emotional cues in speech can significantly enhance the quality of emotional support and intervention provided to hotline users. Tavi et al. analyzed prosodic speech features, such as pitch and rhythm, in suicide emergency calls [15], while Iyer et al. used voice analysis to assess distress in Australian hotlines [16]. The recent advancements in deep learning technologies have facilitated the emergence of Transformer [17]-based models—such as Wav2Vec 2.0 [18], HuBERT [19], and Whisper [20]—as innovative tools for speech feature extraction. These models, trained on large-scale, unsupervised speech data, have demonstrated significant efficacy in a variety of speech-related task. Chen et al. achieved a 76.96% F1 score in recognizing negative sentiment using Wav2Vec 2.0 in Chinese hotline data [21]. Concurrently, Song et al. utilized Whisper for feature extraction, incorporating positional embeddings to assess psychological risk in hotline contexts. Their approach achieved a 71.15% F1 score in predicting future suicides [22].

While acoustic features provide useful insights, they often lack the contextual depth needed to fully capture psychological states. Text, on the other hand, offers richer emotional context. Natural Language Processing (NLP) techniques are particularly effective at analyzing the semantics, syntactic structure, and emotional undertones, all of which are essential for identifying psychological crises. The emergence of large language models (LLMs), such as Bidirectional Encoder Representations from Transformers (BERT) and Generative Pre-trained Transformers (GPT), has transformed the field of NLP [23]. BERT's bidirectional encoding captures complex dependencies [24], while GPT's autoregressive architecture generates coherent, contextually relevant text [25]. GPT-4, fine-tuned with reinforcement learning from human feedback (RLHF), has improved its alignment with human values and safety [26]. Recent advancements, such as DeepSeek's R1 Zero, have introduced capabilities like self-verification, reflection, and extended chains of thought (CoT), marking a significant leap in AI's ability to engage in deeper reasoning. DeepSeek also introduced R1, designed for mathematical, coding, and logical problem-solving to enhance autonomous decision-making. The potential application of LLMs in psychological crisis intervention has sparked significant interest. For example, Salmi et al. used BERTopic models [27] to examine tone and content shifts during the COVID-19 pandemic [28], and Wang et al. applied NLP techniques to extract key sentences from Taiwanese hotline transcripts, achieving a 65.94% F-score for detecting the most dangerous situations [29]. Moreover, Cui et al. demonstrated the efficacy of combining Whisper with LLMs to enhance suicide risk detection, achieving an accuracy of 80.7% and an F1 score of 84.6% [20]. Despite these advances, many studies focus on isolated risk factors, such as suicidal ideation, depression, or recent suicide attempts, with few integrating a holistic analysis of emotional states and suicidal behaviors. This gap underscores the need for advanced methods capable of synthesizing both emotional and behavioral dimensions, which could significantly improve intervention strategies.

In this study, we analyzed 1,057 calls from the Hangzhou Psychological Support Hotline at Hangzhou Seventh People's Hospital between 2022 and 2023, including 526 high-risk calls, and the study design for crisis detection is shown in Fig 1. Our first key contribution is applying pre-trained models to the domain of psychological crisis hotline risk assessment. We explore the processing of audio and transcribed text from hotlines to predict callers' emotional state and suicidal behavior. To assess clinical relevance beyond standard accuracy metrics, we pilot a human expert evaluation framework. This framework evaluates GPT-generated explanations based on consistency with labeling, plausibility of reasoning, completeness and factual accuracy, clinical and situational relevance, as well as clarity and comprehensibility. Our second contribution is the development and validation of a deep learning model using GPT-embeddings for multi-label prediction,

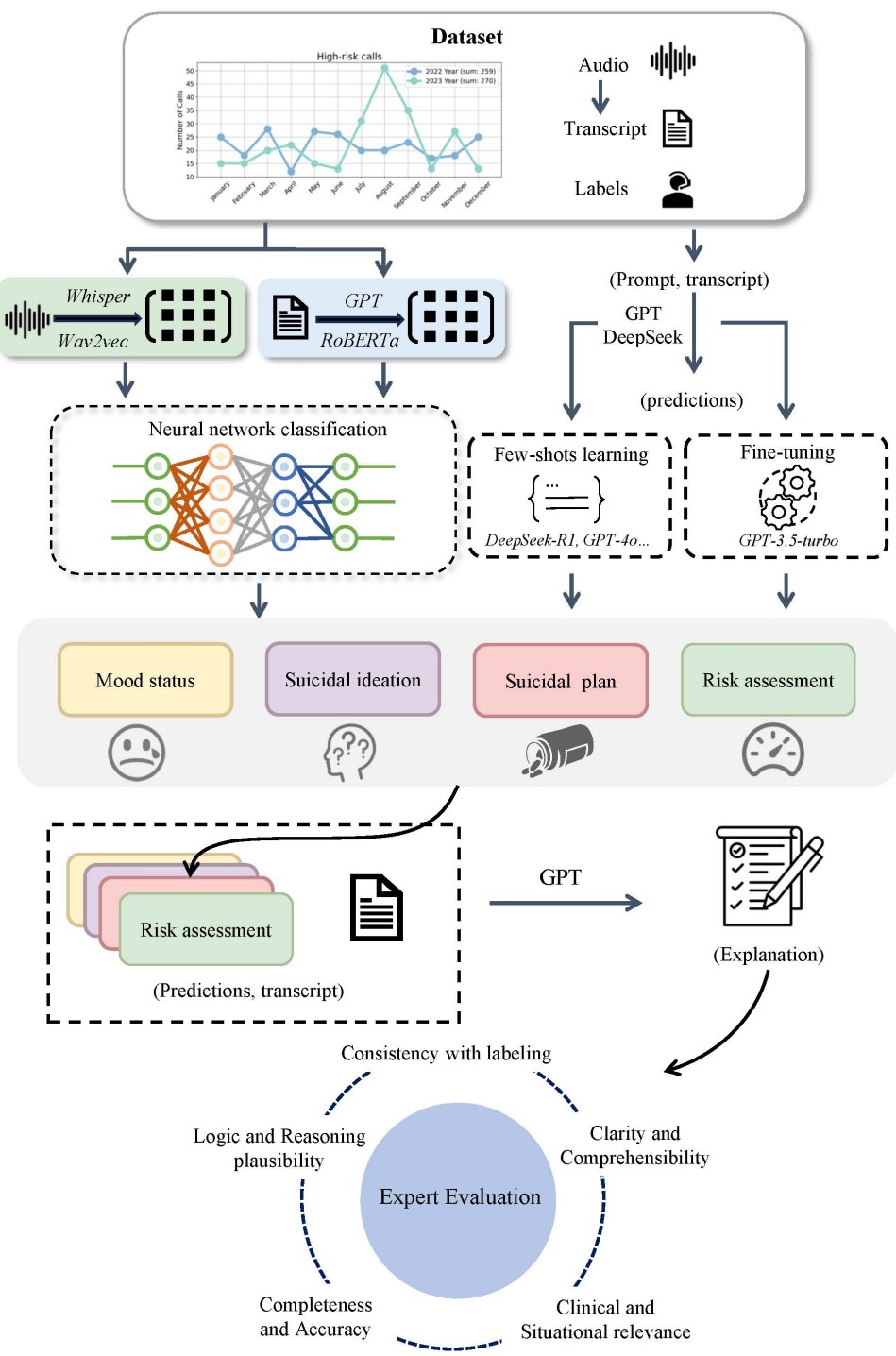

**Fig 1. An overall illustration of our framework.** The framework diagram shows the process and methods of psychological support hotline data analysis. We collected high-risk call data from the Hangzhou psychological support hotline between 2022 and 2023. The aim was to explore methods for automatically and effectively detecting psychological crises using audio and transcribed text from hotlines. To achieve this, two strategies were employed for a multi-dimensional prediction task: (1) a deep learning classification framework utilizing pre-trained models, wherein embeddings were first extracted from audio and transcribed texts based on pre-trained models, followed by the design of a neural network that integrates the reconstruction task of an autoencoder with the classification task to enable end-to-end prediction; (2) LLM-based prediction through prompt engineering, which explores the

capability of advanced LLMs to process hotline call texts via few-shot learning or fine-tuning. Subsequently, GPT was used to generate natural language explanations for model's predictions, including emotional state, suicidal ideation, suicide planning, and risk assessment. Finally, an expert evaluation framework was employed to assess the explanations from multiple perspectives, ensuring the reliability and applicability of the model.

achieving an F1 score of 80.48% [80.18%, 80.78%] in identifying high-risk calls. The third contribution is the introduction of a prompt engineering strategy that is guided by preset prompts and does not require large amounts of labeled data for training [30,31]. In particular, DeepSeek-R1 and GPT-4o, leveraging few-shot learning, demonstrated comparable performance to traditional deep learning models reliant on pre-trained embeddings. Finally, we present a detailed analysis of the limitations of LLMs as revealed by human expert evaluation. While our results illustrate the potential of LLMs for multidimensional prediction in psychological hotline settings, they also identify critical gaps that must be addressed to ensure clinical safety and viability in real-world deployments.

## Results

Table 1 lists the characteristics of the dataset from the Hangzhou Psychological Support Hotline, including key demographic and mental health information. The dataset includes call recordings and corresponding transcripts of a total of 1,057 calls, of which 526 were high-risk calls (approximately 50%).

**Table 1. Comparison of basic characteristics.**

| Variable | High risk | Control group | All |
|---|---|---|---|
| **Sex** | | | |
| Male/Female | 249/277 | 257/274 | 506/551 |
| **Age *** | | | |
| Below 19/20–30/Above 30 | 230/189/69 | 229/190/70 | 459/379/139 |
| **Education level *** | | | |
| Low/ Medium/ High | 11/275/89 | 7/268/148 | 18/543/237 |
| **Occupation *** | | | |
| Unemployed/Employed | 106/330 | 71/387 | 177/717 |
| **Marital status *** | | | |
| Married/Single/Divorced | 33/418/16 | 24/448/5 | 57/866/21 |
| **Mood status** | | | |
| Normal/Depression | 68/458 | 509/22 | 577/480 |
| **Suicidal ideation** | | | |
| No/Yes | 26/500 | 422/109 | 448/609 |
| **Suicide plan** | | | |
| No/Yes | 157/369 | 527/4 | 684/373 |

Some variables involved in the study may have missing data, and been marked "*", resulting in varying numbers of callers for each variable, not encompassing the entire number of callers. A complete-case analysis strategy was applied to these variables, with denominators as follows: Age (High-risk: $n = 488$; Control: $n = 489$; Total: $n = 977$), Education level (High-risk: $n = 375$; Control: $n = 423$; Total: $n = 798$), Occupation (High-risk: $n = 436$; Control: $n = 458$; Total: $n = 894$), and Marital status (High-risk: $n = 467$; Control: $n = 477$; Total: $n = 944$).

Education levels: "primary school or below" for individuals with low education, "high school or vocational school" for individuals with medium education, and "college or above" for individuals with high education.

HR: high risk calls, CG: control group, which means non-high-risk calls.

## Demographic and clinical information of the Psychological Support Hotline data

The dataset includes all high-risk calls to the psychological support hotline for mental health issues from January 2022 to December 2023 (2022: 259 cases; 2023: 267 cases) and corresponding matched control group calls. The high-risk and control groups are comparable in demographic characteristics, such as gender and age. Calls from the 2023 high-risk group and matched controls were used as the prospective test set. The high-risk group is predominantly young (230 aged <19 years; 189 aged 20–30 years), with most individuals having medium education (275), 89 with higher education, and 11 with low education. Unemployment was a notable socioeconomic factor, with 107 high-risk callers reporting unemployed status. Marital status data show that most individuals are single (418), with 33 married and 16 divorced. Compared with the control group, depressive status, suicidal ideation, and suicide plan were always significant features in the high-risk group. In summary, we can draw some obvious high-risk characteristics, such as young people, unmarried, unemployed, low education level, and individuals reporting depression, suicidal ideation, or suicide plan.

## Multidimensional prediction with crafted models

Fig 2 shows the performance of a multidimensional deep learning classification model using different pre-trained models in the emotional state, suicidal ideation, suicide plan and high-risk vs. non-high-risk classification tasks. The figure

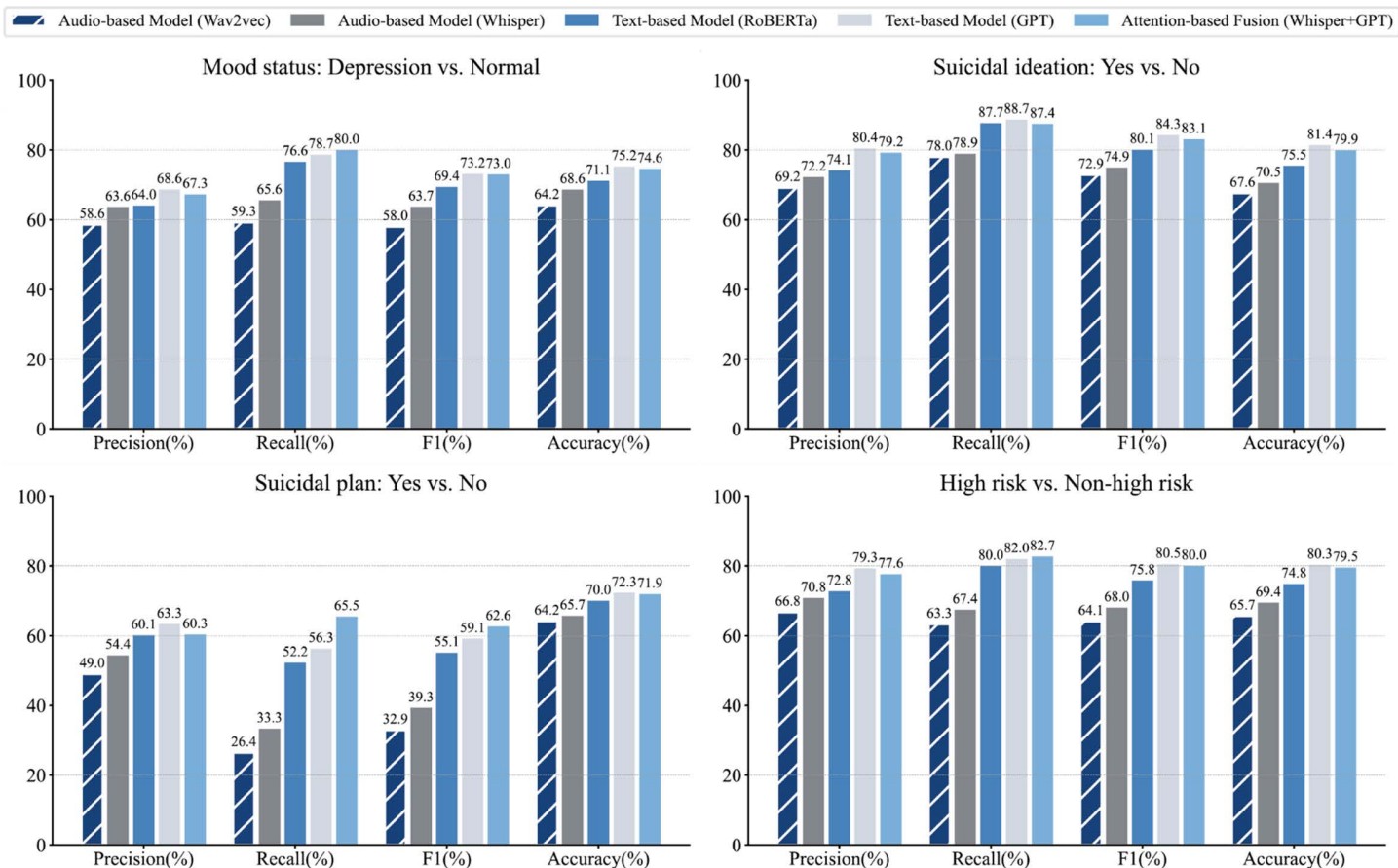

**Fig 2. Performance evaluation for the multidimensional prediction with crafted models.** The figure presents the performance of the multidimensional deep learning classification model using different pre-trained models. The bars present the average value of the model evaluation indicators in the prospective test set after 100 iterations. The percentage is shown above each column. The results compare the classification performance of extracting embeddings from audio and transcribed text using different pre-trained models. Wav2vec and Whisper are pre-trained models applied to process audio, while text embeddings are derived from the RoBERTa and GPT models.

presents the average value of the model evaluation indicators in the prospective test set after 100 iterations. Detailed performance comparisons are provided in S4 Table.

Text-based deep learning frameworks utilizing pre-trained models (RoBERTa and GPT) consistently outperformed audio-based models (Wav2Vec 2.0 and Whisper) across all tasks. Specifically, GPT embedding consistently achieved the highest accuracy and F1-Score across all four classification tasks—Mood Status (Accuracy: 75.25% [75.09%, 75.42%], F1-Score: 73.16% [72.85%, 73.48%]), Suicidal Ideation (Accuracy: 81.36% [81.16%, 81.55%], F1-Score: 84.29% [84.10%, 84.47%]), Suicidal Plan (Accuracy: 72.28% [72.10%, 72.45%], F1-Score: 59.11% [58.19%, 60.00%]), and High Risk vs. Non-High Risk (Accuracy: 80.29% [80.10%, 80.47%], F1-Score: 80.48% [80.18%, 80.78%]). These results indicated that GPT is particularly attuned to the linguistic nuances and contextual cues that are critical for detecting elevated risk levels.

To evaluate model's generalizability to real-world settings where high-risk calls are infrequent, we performed a prevalence-aware simulation. We simulated a 1.5% high-risk prevalence—mirroring 2023 hotline statistics—with a 4:270 high-risk-to-control ratio. Comprehensive performance metrics for each method under these imbalanced conditions are provided in S7 Table. We observed that while precision naturally declined due to low prevalence, GPT embedding still maintained high recall (78.75% [74.50%, 82.76%]) and accuracy (78.67% [76.59%, 80.70%]) in the High Risk vs. Non-High Risk classification task. This suggests the model remains an effective decision-support tool for prioritizing urgent interventions even in imbalanced conditions.

The performance profile of the GPT embedding framework aligned effectively with clinical priorities. For suicidal ideation detection, where sensitivity is paramount to prevent missed cases, the model achieved a high recall of 88.69% [88.06%, 89.32%]. Conversely, for suicide plan detection—which triggers resource-intensive interventions—the results reflected a higher precision of 63.33% [62.92%, 63.77%], thereby helping to mitigate false positives. In contrast, audio-based models showed limited utility. While acoustic features extracted by Wav2Vec 2.0 and Whisper yielded a moderate F1 score of 63.7% [62.51%, 64.65%] for emotional state classification, they significantly underperformed in detecting complex behavioral markers like suicide plans. By adopting Bidirectional Long Short-Term Memory (BiLSTM) strategies to preserve temporal dynamics, we observed a significant improvement in recall for auditory models; however, the corresponding increase in F1 score remained modest (see S5 Table). This indicates that while acoustic properties may signal mood shifts, they lack the semantic resolution required to identify specific suicidal intent.

We further investigated multimodal synergy using an Attention-based Gated Fusion architecture. However, fusion experiments did not yield statistically significant performance gains over the unimodal GPT text baseline. Statistical comparison results are detailed in S4–S7 Figs. In 100 experimental iterations, the learnable attention weights consistently converged to approximately 1 in favor of the text modality. This convergence indicates that the model autonomously identified the linguistic representations as the primary carrier of predictive signal, treating the audio stream as largely redundant relative to the rich semantic features captured by the GPT embeddings.

## Multidimensional prediction with prompt engineering

For multi-label prediction based on prompt engineering (DeepSeek series and OpenAI's GPT models), Fig 3 shows the evaluation results based on 5 repetitions of the same prompt input (S6 Table). Statistical analysis indicated that while performance varied across the four tasks under few-shot learning, the inter-model differences were not statistically significant (S4–S7 Figs).

Notably, DeepSeek-R1 consistently achieved the highest recall across all evaluated tasks. In the High Risk vs. Non-High Risk classification, DeepSeek-R1 and GPT-4o achieved peak accuracies within their respective series at 80.11% [78.99%, 81.12%] and 82.68% [81.86%, 83.46%], effectively matching the performance of state-of-the-art embedding-based deep learning models.

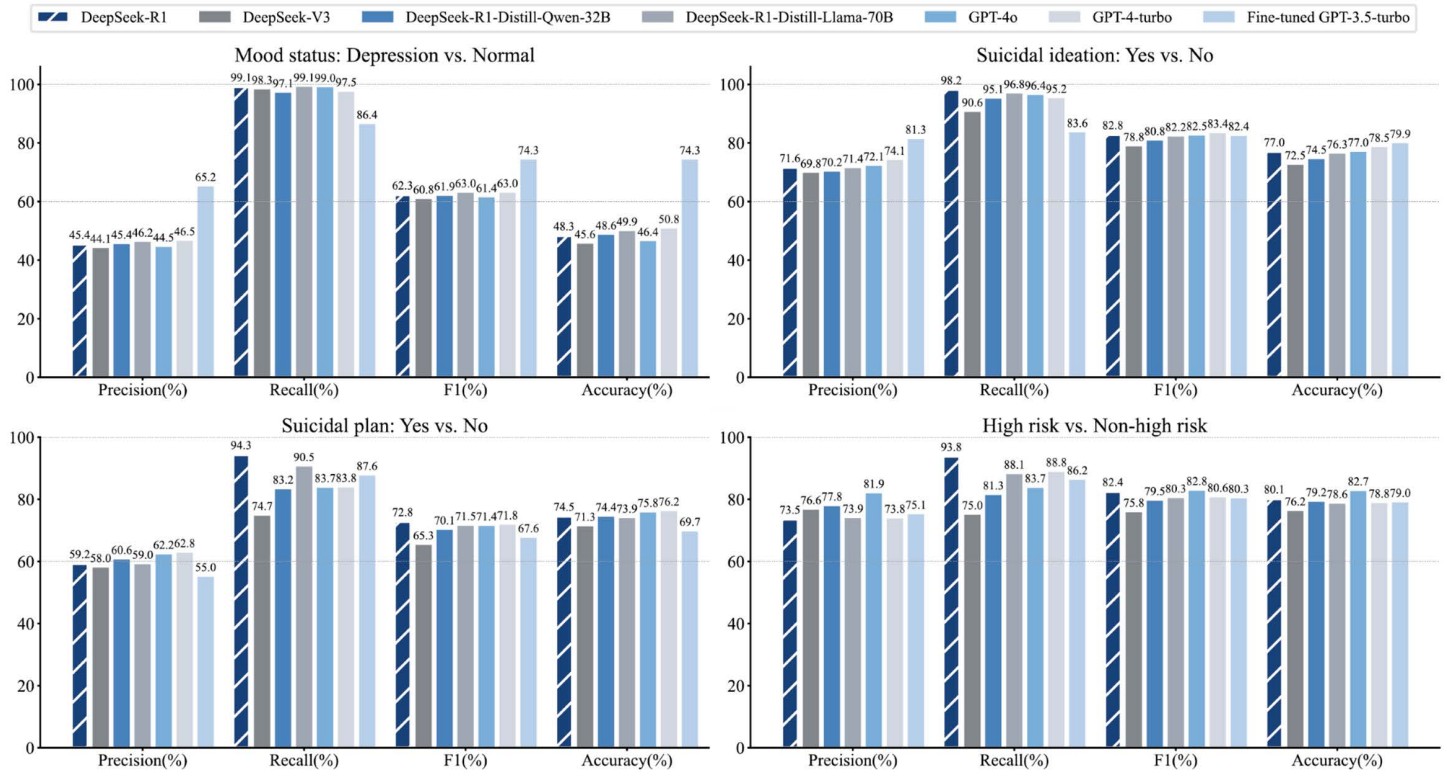

**Fig 3. Performance evaluation for the multidimensional prediction with prompt engineering.** The figure presents the performance of the multi-label prediction based on prompt engineering (DeepSeek series and OpenAI's GPT models). The bars show the evaluation results based on 5 repetitions of the same prompt input. The percentage is shown above each column.

The fine-tuned GPT-3.5-turbo model showed marked improvements in Mood Status classification (Accuracy: 74.26% [73.74%, 74.86%]; F1-score: 74.28% [73.75%, 74.88%]), suggesting that domain-specific training enhances model sensitivity to subtle emotional cues in crisis hotline transcripts. Furthermore, in the Suicidal Plan detection task, DeepSeek-R1, GPT-4o, and GPT-4-turbo all significantly outperformed the best embedding-based deep learning models, underscoring the superior capability of LLMs in complex reasoning and contextual integration.

The DeepSeek-R1-Distill models also delivered strong results, though they fell short of DeepSeek-R1's performance. Nevertheless, the distilled models remained competitive, indicating their potential for various real-world applications where smaller models are beneficial due to computational constraints. Notably, DeepSeek-R1-Distill-Llama-70B surpassed DeepSeek-R1-Distill-Qwen-32B, showing that larger models can yield better results.

Finally, these results demonstrate that LLMs utilizing few-shot learning are highly competitive with traditional deep learning models. However, a key advantage of the GPT Embedding-deep learning model lied in its ability to generate Receiver Operating Characteristic curves (ROC, S3 Fig). This allows for a granular evaluation of the recall-specificity trade-off across varying decision thresholds, providing the operational flexibility necessary to adapt to different clinical intervention scenarios.

## Manual evaluation results

To address the opacity of context embedding, we leveraged the generative capability of GPT to produce human-readable explanations based on its predictions. These explanations extracted key information from the model's inputs and outputs,

enhancing interpretability and transparency. To ensure practical applicability, we involved domain experts to evaluate the explanations, turning technical details into actionable insights and validating the clinical relevance of the model's predictions. Fig 4 displays the evaluation of GPT-generated explanations based on call texts and corresponding model-predicted labels, scored by two experts across different dimensions. Given the subjectivity of scoring, a 1-point difference is considered consistent, and high agreement between experts was demonstrated.

Fig 4 reveals that explanations for correct high-risk predictions scored higher in dimensions like label consistency, logical reasoning, information completeness, and clinical relevance. True positives and true negatives consistently outperformed false positives and false negatives. High scores for correct predictions indicated interpretable, well-reasoned explanations, while low scores for incorrect predictions suggested inadequate justification for erroneous decisions. Although the generated explanations ware rated highly for clarity and comprehensibility, this did not guarantee reliability.

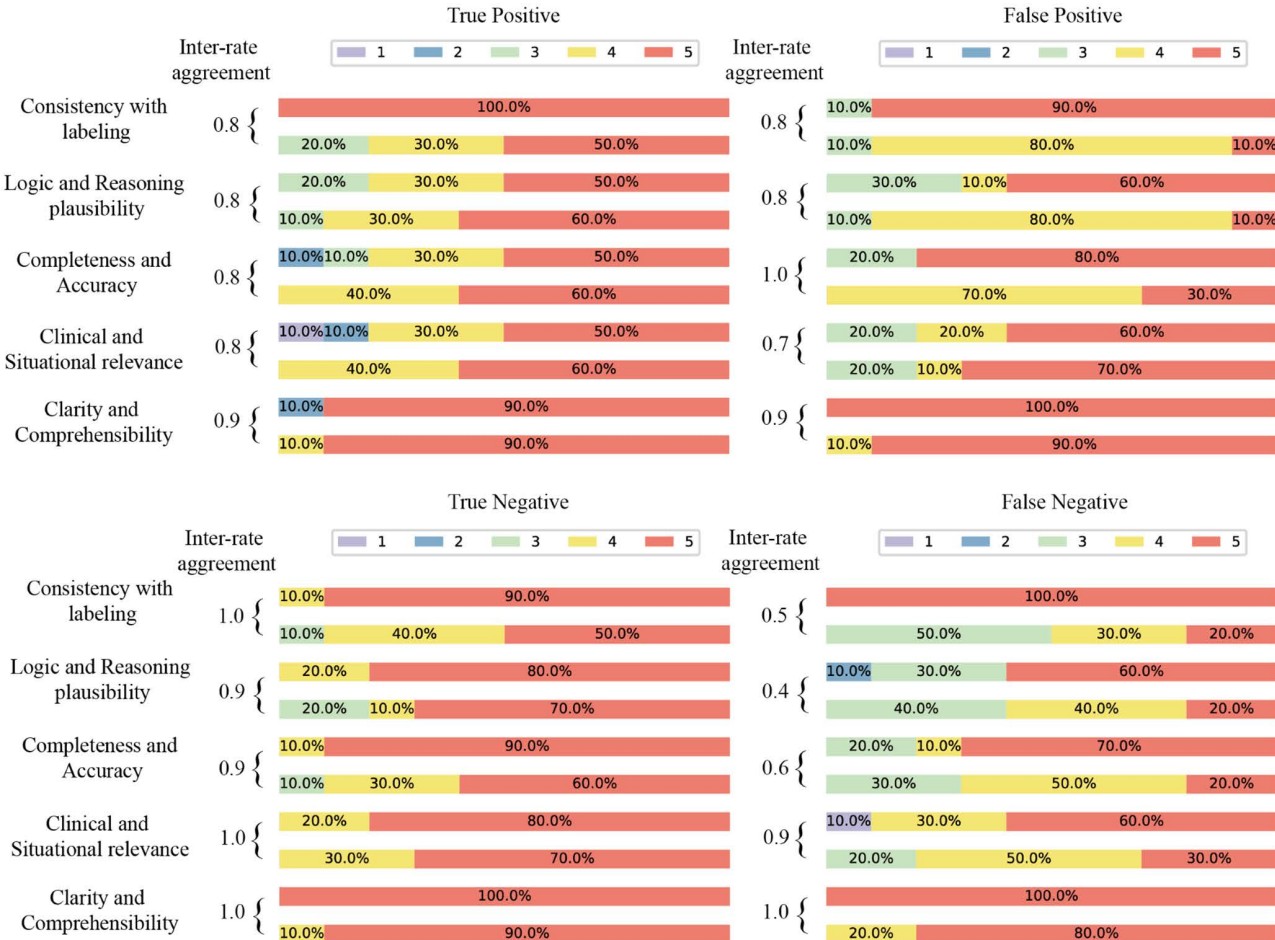

**Fig 4. Results of manual evaluation of generated explanations for four categories under "high-risk vs. non-high-risk" label.** The percentages corresponding to the different scores displays the evaluation of GPT-generated explanations based on call texts and corresponding model-predicted labels, scored by two experts across different dimensions. Given the subjectivity of the scoring process, a score difference of 1 point is considered consistent. True positives, false positives, true negatives, and false negatives refer to the four types of correspondence between the model's predictions and the ground truth within the "high-risk vs. non-high-risk" label. The specific scoring rubrics for these five dimensions, including detailed definitions and clinical criteria, are provided in S1 Appendix.

GPT excels at generating clear and understandable text even when predictions are incorrect, but logical reasoning and factual consistency may be compromised.

Explanation quality indirectly reflects the model's predictive ability in the psychological crisis prediction task. When the model predicts high risk, it can accurately identify key indicators like explicit suicidal thoughts or plans, such as "I don't want to live anymore" or "I can't hold on any longer." However, it occasionally over-relies on negative emotional keywords, misclassifying non-suicidal cases as high-risk. When assessing depression, the model might categorize calls as depressive based on mentions of stress or negative emotions, but it needs a deeper contextual understanding to differentiate between temporary mood swings and long-term depression. Furthermore, subtle or complex emotional expressions were sometimes overlooked, leading to errors. This highlights the need for the model to be more sensitive to emotional fluctuations and potential suicide risks, enhancing its accuracy in challenging cases.

## Discussion

Our study demonstrates the potential of using large-scale pre-trained models, specifically GPT-based models, to build a deep learning framework for the multi-label prediction of emotional states and suicidal behavior in hotline calls. As digital tools become increasingly integral to mental health support [32], this research highlights the viability of automating suicide risk assessments using pre-trained models. Given the accessibility and cost-effectiveness of hotlines [12], particularly in resource-limited settings, the findings contribute to the growing field of mental health technology by suggesting that automated systems could enhance the efficacy of human responders.

Hotlines have long served as a vital lifeline for individuals in crisis [12], offering immediate access to help, particularly in regions with scarce mental health resources. Automated tools for suicide risk assessment present an opportunity to augment traditional strategies by prioritizing callers in urgent need, enabling more efficient resource allocation [5]. Our model, developed using large-scale hotline data, illustrates the promise of deep learning and LLMs in refining suicide risk assessments, potentially providing an additional layer of support for responders.

The study compared two distinct strategies—1) deep learning classification framework with pre-trained models and 2) LLM-based prediction via prompt engineering—demonstrating the superiority of GPT embedding in combination with deep learning framework for emotional and suicidal behavior classification. This hybrid approach yielded highly accurate and stable results, suggesting that integrating state-of-the-art language models with traditional deep learning architectures can enhance the reliability of automated suicide risk assessments. Text-based models, such as GPT and RoBERTa, outperformed audio models like Wav2Vec 2.0 and Whisper, as they capture complex linguistic structures and emotional semantics more effectively. While adopting BiLSTM strategies to preserve time-series information significantly improved acoustic-based recall, the overall performance ceiling remained firmly dictated by the text modality. While affective computing theory posits paralinguistic features—such as prosody, rhythm, and tone—are indispensable for emotion recognition [33], our fusion experiments revealed a distinct modality dominance of text in the context of crisis detection.

The model's negligible reliance on audio cues can be attributed to the distinctive characteristics of hotline interactions. In this setting, a distressed or depressive tone acts as a pervasive baseline across nearly all callers, making acoustic features a poor discriminator for distinguishing immediate lethality from general distress. Conversely, markers of high-risk ideation are predominantly semantic—encoded in the propositional content of callers' disclosures (e.g., concrete plans or explicit intent) rather than in their vocal delivery. Furthermore, audio data is inherently vulnerable to environmental noise, recording variability, and speaker-specific characteristics (e.g., accent, pitch), all of which tend to introduce between-speaker variability and artifacts. The finding that a text-only modality outperforms multimodal approaches therefore carries important practical implications for mental health interventions. It demonstrates the sufficiency of GPT-based systems in settings where audio data is unavailable, unreliable, or ethically sensitive. This, in turn, enables the deployment of high-precision risk detection models in online text-based counseling services, a modality that affords greater privacy and is increasingly favored by adolescents who may be reluctant to engage in voice calls [34,35].

In contrast, GPT-4o and DeepSeek-R1, employing few-shot learning, demonstrated performance comparable to the GPT-embedding deep learning model across multiple tasks, thus confirming the efficacy of leveraging LLMs in automated crisis detection in hotline contexts. Specifically, in tasks involving suicide planning and high-risk prediction, DeepSeek-R1 exhibits robust reasoning capabilities, maintaining stable classification even with limited samples. Furthermore, fine-tuning GPT-3.5-turbo model yields significant improvements in emotion recognition tasks, addressing its bias toward negative emotions and enhancing detection of subtle emotional shifts.

While our study utilized a matched case-control design to optimize feature learning, this may not fully reflect model performance in real-world settings where high-risk callers represent a small minority (approximately 1.5%). Nevertheless, our prevalence-aware simulation demonstrates that the GPT-embedding deep learning model remains a robust screening instrument. Although its Positive Predictive Value (PPV) is sensitive to such skewed distributions, the model maintains a high recall. In the high-stakes context of psychological crisis intervention, prioritizing recall is paramount to ensuring that no high-risk individual is overlooked. The high sensitivity of this framework suggests its potential to function as an automated "safety net", effectively alerting operators to latent crises that necessitate immediate clinical attention.

Furthermore, we extended our evaluation to the Distress Analysis Interview Corpus (DAIC) [36] to assess the framework's cross-cultural robustness. By extracting GPT-embeddings from DAIC transcripts, a simple SVM classifier achieved a classification accuracy of 78.72% and a recall of 71.43% for depression detection (S8 Fig). These preliminary results indicate that GPT embedding model successfully captures universal linguistic markers of psychological distress within a culturally diverse population.

While the GPT-embedding deep learning model achieved promising results in classification accuracy, challenges in interpretability remain. While LLMs such as GPT can generate human-readable explanations for the model's predictions that go beyond simple keywords and provide a more comprehensive and coherent understanding of the model's decision-making process, these explanations can suffer from oversimplification, insufficient details, or incoherent reasoning. Clinical experts evaluated the explanations, noting their potential applicability but also identifying issues with logical reasoning and factual accuracy. In some cases, the generated explanations may appear coherent but are factually incorrect or lack sufficient justification for the model's decisions [37]. This issue is particularly concerning in high-stakes scenarios like suicide risk assessment, where misinterpretations may lead to inappropriate responses or undermine trust in the system. Future work must address the gap between generating plausible explanations and ensuring their factual accuracy and logical consistency. Our expert evaluation framework represents a promising pilot approach, but our chosen axes of evaluation were not exhaustive and were subjective in nature. A critical long-term goal for this field is the development of broader, automated, and standardized objective benchmarks. Such standardized frameworks are essential to move beyond subjective expert reviews and will be instrumental in further enhancing the clinical safety, transparency, and overall trustworthiness of LLM-generated insights in mental health services.

This study has several limitations. First, the process of labeling emotional data and risk levels is inherently subjective, which may introduce inconsistencies or errors in the ground truth. Although we relied on the systematic logs of trained operators following standard protocols, the absence of an objective biological "gold standard" for psychological crises means that labels rely heavily on clinical judgment. This subjectivity may limit the model's ability to distinguish subtle nuances in borderline cases. Additionally, audio-text fusion experiments did not yield statistically significant performance improvements. Future work should move beyond modality-specific temporal modeling toward fine-grained, cross-modal attention mechanisms. Such strategies could enable the model to dynamically align transient acoustic markers with specific high-risk linguistic segments, thereby enhancing the synergy between vocal urgency and articulated intent. Third, while our prevalence-aware simulation validated the GPT-embedding deep learning model as a high-recall screening tool, its PPV remains sensitive to the low-prevalence typical of real-world settings. Future deployment will necessitate cost-sensitive learning or advanced data augmentation to manage extreme class imbalances without compromising clinical utility. Fourth, while we utilized a high-performance ASR system, the transcripts did not undergo manual correction. Although

LLMs are robust to ASR noise, future work involving fine-grained linguistic analysis may benefit from human-in-the-loop correction. Finally, to further improve generalizability, future studies should incorporate multicenter validation across diverse hotlines and populations to assess model robustness in varied cultural and clinical contexts. Additionally, longitudinal follow-up is essential to evaluate the model's predictive value over time by linking call data to subsequent mental health outcomes.

In conclusion, this study establishes that deep learning models utilizing pre-trained text embeddings can accurately identify high-risk callers in psychological support hotlines, significantly outperforming audio-based approaches. The findings challenge the necessity of multimodal analysis in hotline settings, demonstrating that semantic content is the primary driver of risk detection. Furthermore, LLMs demonstrated comparable efficacy through few-shot learning, offering a flexible, low-resource alternative to traditional training. Ultimately, these automated systems show great promise not as replacements for human judgment, but as sophisticated decision-support tools capable of enhancing the speed and accuracy of life-saving interventions in psychological support hotlines.

## Materials and methods

### Data collection and selection

This study is based on data from the Hangzhou Psychological Support Hotline, which provides free 24-hour services nationwide, offering psychological support to tens of thousands of callers annually. All hotline operators undergo suicide risk assessment training before answering calls. Operators collect basic demographic information from callers (such as gender, age, education level, marital status, occupation, etc.) as general demographic data. In addition, during the consultation process, callers' emotional states, suicide risks, and related factors are assessed.

This study included all high-risk calls to the psychological counseling hotline for mental health issues from January 2022 to December 2023 (2022: 259 cases; 2023: 267 cases).

High-risk call evaluation criteria (for details, see S1 Table):

Definition of high-risk calls [38]: (1) Individuals with suicidal thoughts and specific suicide plans. (2) Individuals with suicidal thoughts but no plan, with a risk factor score≥5.

In addition, the emotional state of the callers is also considered as a risk factor in the scoring system (Moderate depression is assigned 1 point, and severe depression is assigned 2 points).

The Depression and Suicide Risk assessment questionnaire (S1 Table) was administered via a semi-structured approach. To maintain working alliance, operators did not read the questionnaire as a rigid script. Instead, they used the items as a systematic framework to guide the conversation, probing for specific risk factors as they naturally emerged. Following each call, operators utilized their clinical judgment and the caller's disclosures to complete a standardized assessment log, which served as the basis for the labels used in this study.

Based on the demographic information of high-risk calls, non-high-risk calls that matched the call time, gender, marital status, and call type of high-risk calls were selected as the control group with the call duration, call date, and the closest call to the high-risk call.

Exclusion criteria were:

(1) invalid calls (lasting less than 60 seconds, silent or harassing calls);

(2) seeking information only, i.e., not seeking psychological help;

(3) repeated calls. For repeated calls, only the first call that was fully assessed was included in the data analysis.

A total of 1,057 call recordings were included in the study. The recordings were captured in a dual-channel format, with the caller and the operator recorded on independent tracks. By specifically extracting the caller's channel (Left), we achieved deterministic speaker isolation. The caller's audio was then preprocessed through removing silence periods,

denoising, and loudness normalization. We used the Voice Activity Detection (VAD) model in the pyannote.audio library to eliminate silence periods in the recording. The recordings were also converted into text using iFlytek's speech transcription API (https://www.xfyun.cn/doc/asr/ifasr_new/API.html), resulting in a high-fidelity textual corpus for subsequent modeling. The transcriptions included the conversations between the operator and the caller, all in the form of dialogue. In this study, we only focus on the text content of the caller. Based on the operators' post-call annotations, four binary labels were established for each recording: mood status, suicidal ideation, suicidal plan, and high-risk status. For mood status, we merged three levels (mild, moderate, and severe depression) into a single positive "depression" label. These labels exhibit significant internal interdependencies; for instance, "depressed state", "suicidal ideation", "suicidal plan", and "high risk" often appear at the same time (for details, see S1 Fig).

## Ethics statement

The investigation was approved by the Ethics Committee of Hangzhou Seventh People's Hospital (2025–054). On the one hand, the hotline data in this study are recorded, and callers are informed before collecting information. Data collection can only take place with the informed consent of the caller. On the other hand, the hotline itself is anonymous, and this study is retrospective and does not involve any sensitive personal information, so the Ethics Committee of Hangzhou Seventh People's Hospital concluded that the study had no ethical implications.

## Deep learning prediction framework with pre-trained model

Computers cannot process audio or text the way humans do; they require numerical data as input. In the fields of speech signal processing and NLP, audio or text is converted into high-level vector representations, which are then fed into machine learning and deep learning algorithms. The quality of these embeddings has a significant impact on machine comprehension; better embeddings can lead to enhanced performance. This study constructs a multi-dimensional classification model using audio and transcribed text from hotlines, leveraging large-scale pre-trained models. By utilizing pre-trained models, we effectively extracted meaningful features from raw hotline data and trained neural network classification models for multi-label binary classification tasks, including identifying emotional states, suicidal ideation, suicidal behavior, and high-risk indicators. For the audio modality, pre-trained models such as Wav2vec 2.0 [18] and Whisper [39] are employed to capture acoustic features and speech patterns. For the text modality, we select RoBERTa [40] and GPT [25] to generate contextual embeddings. We conducted a comparative analysis using different pre-trained models. The details are as follows.

Wav2Vec 2.0 [18] is a self-supervised learning framework developed by Facebook AI, specifically designed for building automatic speech recognition (ASR) systems. This framework learns feature representations directly from raw audio data. It encodes audio using a multi-layer convolutional neural network and then masks spans of the generated latent speech representations, similar to the training methods used in language models like BERT [24]. In our experiments, we used the pre-trained model version wav2vec2-large-xlsr-53-chinese-zh-cn, which has been fine-tuned on several Chinese speech datasets, including Common Voice 6.1 [41], CSS10 [42], and ST-CMDS [43]. This fine-tuning enhances its relevance and effectiveness for Chinese speech characteristics. To process hotline audio, each call is first segmented into 10-second clips, which makes processing more manageable while maintaining conversation context. These 10-second segments are passed through Wav2Vec 2.0 to generate high-dimensional embeddings that capture underlying acoustic patterns. First, an averaging operation was applied to these embeddings to form a global representation of the call. However, to better capture the paralinguistic dynamics and emotional evolution throughout the consultation, we further implemented a temporal modeling architecture using a BiLSTM network. The sequence of segment embeddings is fed into the BiLSTM layers to extract dependencies from both forward and backward temporal directions, capturing critical nuances such as hesitation patterns or distress escalation. The final hidden states are aggregated via a pooling layer to generate a comprehensive temporal audio representation for downstream multi-label classification.

Whisper is a pre-trained model designed for ASR and speech translation, which is trained on 680,000 hours of multi-language and multi-task supervised data collected from the web [39]. Whisper's architecture is an end-to-end encoder-decoder Transformer. Whisper demonstrates strong zero-shot performance, enabling it to generalize across many datasets and domains without fine-tuning. In our experiments, we used four versions of Whisper, including Whisper-small-Chinese-base, Whisper-small, Whisper-medium, and Whisper-large-v3. Similar to Wav2Vec 2.0, when using Whisper to extract embeddings, the hotline audio is also first segmented into 10 seconds.

RoBERTa (Robustly Optimized BERT Approach) is a variant of BERT that improves the model's performance by increasing the amount of training data, extending training time, and using dynamic masking, among other methods. "WWM" stands for Whole Word Masking, which means that a whole-word masking strategy was used during pre-training, enabling the model to better capture the semantic information of the Chinese language. We used the Chinese pre-trained model Chinese-roberta-wwm-ext [40] to create text embeddings for the hotline call transcripts. In our hotline call transcript dataset, the text length is variable. To ensure consistent and effective processing of text segments with different lengths, we used the hidden state of the special [CLS] token as a global representation of the text, capturing its overall meaning and feeding it into a CNN for classification.

GPT embeddings stand out for their flexible length, adapting to various text sizes and enhancing their versatility across different language processing tasks. These embeddings are generated from proprietary models hosted on OpenAI servers, which are not publicly accessible. We used the OpenAI API function "openai.Embedding.create" to generate GPT embeddings for hotline call transcripts (https://platform.openai.com). The text-embedding-3-large model processes each call transcript and generates text embeddings with a vector dimension of 3,072. The text-embedding-3-large model is based on the GPT architecture, trained using an Autoregressive Language Model (ARLM), predicting the next word in sequence. When processing text, the model typically integrates global contextual information to generate embedding representations. GPT embeddings have a limit of up to 8,191 tokens, which helps in handling larger documents.

After obtaining the embeddings, we designed a neural network classification framework for multi-dimensional prediction tasks, including identifying emotional states, suicidal ideation, suicidal behavior, and high-risk indicators. By combining the reconstruction task of an autoencoder with the classification task, we maximized the utilization of input data features for multi-label classification. This joint learning approach ensures that the extracted features are not only discriminative for classification but also representative of the original data structure, providing a regularization effect that prevents the model from overfitting on high-dimensional noise. The reconstruction task of the autoencoder aims to learn the latent structural features of the data by transforming the input text embedding vector $\epsilon$ into a latent variable $z$ through the Encoder g, and then reconstructing the latent variable back into the original input $\hat{\epsilon}$ through the Decoder d. The goal of the decoder is to accurately reproduce the original input, and the mean squared error (MSE) is used to measure the difference between the decoder's output $\hat{\epsilon}$ and the original input $\epsilon$.

$$L_{rec} = \frac{1}{B} \sum_{i=1}^{B} \sum_{d=1}^{D} (\hat{\epsilon}_i^d - \epsilon_i^d)^2 \tag{1}$$

where $B$ represents the batch size, i.e., the number of input samples processed in one training iteration, and $D$ represents the dimension of each embedding vector. $\epsilon_i^d$ represents the original input value of the $d^{th}$ dimension of the embedding vector for the $i^{th}$ sample.

The classification loss function uses binary cross-entropy loss to measure the difference between the predicted probability $\hat{y}_i^{(j)}$ for each label and the true label $y_i^{(j)}$:

$$L_{clf}^{(j)} = -\frac{1}{B} \sum_{i=1}^{B} [y_i^{(j)} \log\left(\hat{y}_i^{(j)}\right) + (1 - y_i^{(j)})\log(1 - \hat{y}_i^{(j)})] \tag{2}$$

where B is the batch size for each training iteration, and $j$ represents the $j^{th}$ label. The total classification loss $L_{clf}$ is the sum of the losses for the four classification tasks, given by:

$$L_{clf} = \sum_{j=1}^{4} \gamma_j \cdot L_{clf}^{(j)}$$

(3)

where $\gamma_j$ is the weight for the $j^{th}$ classification task, set as $\gamma_1 = \gamma_2 = \gamma_3 = \gamma_4 = 1$ during training.

Finally, the total loss function $L$ combines the losses from the autoencoder reconstruction task and the classification task. The coefficients $\lambda_1$ and $\lambda_2$ are introduced to balance their contributions, as follows:

$$L = \lambda_1 \cdot L_{rec} + \lambda_2 \cdot L_{clf}$$

(4)

By adjusting the values of $\lambda_1$ and $\lambda_2$, the model can flexibly control its focus on the reconstruction and classification tasks, optimizing overall performance.

To investigate the complementary nature of linguistic and acoustic features, we further implemented an Attention-based Gated Fusion architecture. Unlike static concatenation, this approach employs a learnable gated unit (consisting of an MLP with Sigmoid activation) to dynamically compute a scalar attention weight, $\alpha \in [0, 1]$, based on the latent representations extracted from the text encoder and the audio encoder. The final fused representation is derived via a weighted sum:

$$z_{fused} = \alpha \cdot z_{text} + (1 - \alpha) \cdot z_{audio}$$

(5)

This mechanism allows the model to adaptively prioritize the more informative modality while suppressing noise. More details on neural network are provided in the S2 Fig.

## LLM-based prediction via prompt engineering

LLMs are trained unsupervised on large-scale text datasets. With tens of billions or even hundreds of billions of training parameters, they can process and understand a variety of language structures and contexts. In this study, we adopted a prompt engineering approach, utilizing a series of advanced LLMs, including the DeepSeek series and OpenAI's GPT-4, to process psychological support hotline call text analysis. Unlike traditional methods of training deep learning classifiers, prompt engineering does not require a large amount of labeled data for supervised learning. Instead, it uses the generalization ability of pre-trained models and achieves deep analysis of text through clever prompt design [44]. With carefully designed prompts, the model can effectively identify and annotate key mental health features in call transcripts. Prompt engineering allows us to adjust and optimize the model's output by creating and improving input prompts without retraining the model. We used few-shot in-context learning [25] to provide clear task instructions and reference examples in the prompts, including call transcripts and expected responses, to guide the model to the required JSON output format, thereby making predictions on four dimensions: emotional states, suicidal ideation, suicidal behavior, and high-risk status. To enhance inferential rigor, we utilized CoT prompting [30] by incorporating a 7-level risk hierarchy as a reasoning scaffold. This hierarchy decomposes the "high-risk" logical criteria—which involve ideation, planning, and specific risk scores—into a progressive sequence of clinical indicators. This approach assists LLMs in performing explicit reasoning over the same clinical dimensions used in manual labeling while ultimately producing a binary output suitable for rapid crisis prioritization. The full prompt configurations are provided in S2 Table.

For our task, we evaluated OpenAI's GPT-4 series models (GPT-4o and GPT-4-turbo) and the DeepSeek series models (DeepSeek-R1, DeepSeek-V3, DeepSeek-R1-Distill-Qwen-32B, DeepSeek-R1-Distill-Llama-70B). GPT is a state-of-the-art autoregressive language model. Trained on a wide range of Internet texts, it is able to capture the intricate nuances

of language, including sentiment, context, etc. [45], which enables it to better capture subtle sentiment changes and complex language expressions when processing psychological support hotline call text analysis. GPT-4-turbo is the standard version of the GPT-4 model, which focuses on providing high-quality and accurate text generation capabilities. GPT-4o is an optimized version that provides faster speed and lower cost, suitable for scenarios that require cost-effectiveness and fast response. DeepSeek-R1 [46], introduces self-reflection capabilities through reinforcement learning, demonstrating exceptional performance in tasks requiring mathematical reasoning, coding, and complex problem-solving. The DeepSeek-R1-Distill-Qwen-32B and DeepSeek-R1-Distill-Llama-70B are distilled versions of DeepSeek-R1, fine-tuned using samples generated by DeepSeek-R1. Based on the innovative mixture-of-experts (MoE) architecture, DeepSeek-V3 [47] achieves state-of-the-art performance across a variety of benchmarks while maintaining efficient inference, with 671B total parameters.

One limitation of few-shot learning is the randomness and limitations of the provided examples. Fine-tuning is a key process in the GPT API that involves further training pre-trained language models specialized in specific domains or tasks by leveraging domain-specific or task-specific datasets [25]. The fine-tuning process can improve few-shot learning by training on more examples than in the prompts. When the model is fine-tuned, examples do not need to be provided in the prompts, which can save costs and speed up response times. We adjusted the training and validation sets, randomly selected 50 calls from the included 2022 calls, used system prompts, call texts of the selected calls, and corresponding labels (including emotional state, suicidal ideation, suicidal behavior, and High risk vs. Non-high risk) to prepare the training data required for fine-tuning, and converted binary labels to text. Due to the restrictions on model opening, we chose gpt-3.5-turbo-0125 as the pre-trained model for the fine-tuning process.

## Explanation generation and human expert's evaluation

The use of contextual embeddings or audio embeddings, poses challenges for the interpretability of machine learning model decisions. Due to the irreversibility of embeddings, interpretability techniques such as Local Interpretable Model Agnostic Explanations (LIME) [48] or Shapley Additive Explanations (SHAP) [49] are difficult to implement. We propose to leverage GPT's powerful text generation capabilities to generate understandable explanations based on the input text and the model's predicted labels, locate relevant sentences, and provide human-readable explanations for hotline mental health crisis predictions. Here, we designed an additional prompt to extract natural language explanations of the predictions of caller emotional state and suicide risk based on context embedding and neural network models to enhance the transparency of model predictions. The designed prompt is provided in the S3 Table.

To further evaluate the rationality and clinical applicability of the machine learning model for the prediction of caller emotional state and suicide risk, we conducted further manual evaluation on the generated explanations. The correspondence between the model predictions and the ground truth is analyzed on the "high-risk or not" label dimension, which has four categories: true positive (TP), false positive (FP), true negative (TN), and false negative (FN). Ten call texts were selected from each of the four categories as analysis objects. Experts were asked to evaluate and score the explanations generated by GPT based on the call texts and the labels predicted by the corresponding models. The generated explanations were evaluated from the following five evaluation dimensions: consistency with labeling, logic and reasoning plausibility, completeness and accuracy, clinical and situational relevance, clarity and comprehensibility (using a standardized rubric detailed in S1 Appendix), and scored based on the Likert scale (1–5 points). For false positive and false negative samples, analyzing the reasons behind the model's misclassification can help identify and correct potential problems and optimize the reliability and safety of the model in high-risk situations.

## Model evaluation

In our dataset, the data was divided into a training set, validation set and test set. The training set and validation set were used for model training and parameter adjustment, while the test set was reserved for the final prospective evaluation.

The training and validation sets consisted of calls made between January 1, 2022, and December 31, 2022 (randomly split 9:1 between the training set and the validation set). The prospective testing set included calls between January 1, 2023 and December 31, 2023. We utilized large-scale pre-trained models to extract embedding representations of the dataset from audio and transcribed text respectively, and deployed the designed deep learning multi-label classification model on the training set and validation set. To ensure statistical robustness and evaluate model stability under varying data distributions, the training and validation process for deep learning models was repeated for 100 iterations. For each iteration, the training/validation sets were randomly re-partitioned, and the final model was evaluated on the held-out prospective test set. For the LLM-based predictions, considering the inherent stochasticity of the outputs, we repeatedly evaluated the prompt engineering framework on the test set five times. The accuracy of identifying crisis callers in psychological support hotlines must be as high as possible and the performance of the model needed to simultaneously prioritize both precision and recall to mitigate Type I errors (false positives) and Type II errors (false negatives) [50]. Therefore, for our multi-label classification task, we employed precision, recall, F1 score, and accuracy as evaluation metrics for each individual label [51,52]. The overall performance indicators were obtained by aggregating results across all 100 iterations (for deep learning models) and 5 iterations (for LLMs), with results presented as means and 95% Confidence Intervals derived via the Bootstrap method. Comprehensive performance matrices are provided in S4 Table and S6 Table. Furthermore, we implemented a rigorous statistical comparison framework: Welch's t-test was used for large-scale (100 vs. 100) model comparisons; Permutation tests were employed for mixed-scale (100 vs. 5) comparisons; and the Wilcoxon Rank Sum test was applied for small-scale (5 vs. 5) comparisons. To mitigate the risk of Type I errors from multiple comparisons, all p-values were adjusted using the Holm-Bonferroni correction. Comprehensive statistical results are detailed in S4–S7 Figs.

## Model implementation

We use Python as the primary programming language for GPT-based models, conducting experiments with the OpenAI API and the GPT command-line interface (CLI). We use Keras 2.13.1 to define neural network models, train and evaluate the models, and TensorFlow 2.13.1 as the backend to accelerate model computations. The experiments were conducted on a machine equipped with an Intel Xeon Gold 6330 CPU and an NVIDIA A800 GPU. We employed the Adam optimizer with a learning rate set to 1e-4, a batch size of 32, and trained for 150 epochs. The coefficients $\lambda_1$ and $\lambda_2$ were set to 1 and 0.8, respectively. After each epoch, we saved model checkpoints and recorded the training and validation loss and accuracy. If the validation loss did not improve for 40 consecutive epochs, the training would automatically stop to prevent overfitting.

## Code availability

Our code is publicly available at the following link: https://github.com/shuyingrao/Cisis-Detection-in-Psychological-Support-Hotlines.

## Supporting information

**S1 Appendix. Human expert evaluation dimensions.**
(DOCX)

**S1 Fig. Statistics information of high-risk and control group calls in 2022 and 2023.**
(DOCX)

**S2 Fig. The neural network classification framework.**
(DOCX)

**S3 Fig. ROC Curves for the multidimensional prediction with crafted models.**
(DOCX)

**S4 Fig. Statistical Comparison of Multi-Dimensional Prediction Methods Based on Precision.**
(DOCX)

**S5 Fig. Statistical Comparison of Multi-Dimensional Prediction Methods Based on Recall.**
(DOCX)

**S6 Fig. Statistical Comparison of Multi-Dimensional Prediction Methods Based on F1-score.**
(DOCX)

**S7 Fig. Statistical Comparison of Multi-Dimensional Prediction Methods Based on Accuracy.**
(DOCX)

**S8 Fig. Cross-linguistic generalizability validation using the Distress Analysis Interview Corpus.**
(DOCX)

**S1 Table. Depression and Suicide Risk Assessment Questionnaire.**
(DOCX)

**S2 Table. The carefully crafted prompt for the multi-label prediction.**
(DOCX)

**S3 Table. Based on the following prompt, we asked for the reasoning behind the classification.**
(DOCX)

**S4 Table. Performance evaluation for the multidimensional prediction with crafted models.**
(DOCX)

**S5 Table. Performance evaluation for the multidimensional prediction with auditory temporal modeling.**
(DOCX)

**S6 Table. Performance evaluation for the multidimensional prediction with prompt engineering.**
(DOCX)

**S7 Table. Performance evaluation of multidimensional prediction under a prevalence-aware simulation.**
(DOCX)

## Author contributions

**Conceptualization:** Yaoyun Zhang, Haiteng Jiang.

**Data curation:** Shuying Rao, Haidong Song, Tao Li.

**Formal analysis:** Shuying Rao.

**Investigation:** Shuying Rao, Guifeng Deng.

**Methodology:** Shuying Rao, Guifeng Deng, Yaoyun Zhang, Haiteng Jiang.

**Supervision:** Haidong Song, Qiuxia Chen, Mingjin Luo, Yaoyun Zhang, Sha Zhao, Gang Pan, Tao Li.

**Validation:** Qiuxia Chen, Mingjin Luo.

**Writing – original draft:** Shuying Rao.

**Writing – review & editing:** Yaoyun Zhang, Haiteng Jiang.

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
