## [Decision Letter · Decision Letter 0]

5 Feb 2026

PDIG-D-25-01203Automating Multi-label Crisis Detection in Psychological Support Hotlines with Pre-trained ModelsPLOS Digital Health Dear Dr. Jiang, Thank you for submitting your manuscript to PLOS Digital Health. After careful consideration, we feel that it has merit but does not fully meet PLOS Digital Health's publication criteria as it currently stands. Therefore, we invite you to submit a revised version of the manuscript that addresses the points raised during the review process. In addition to the comments provided by external reviewers, there are also comments provided by the section editor (see the 'additional editor comments' section below).   Please submit your revised manuscript by Apr 06 2026 11:59PM. If you will need more time than this to complete your revisions, please reply to this message or contact the journal office at digitalhealth@plos.org.  Please include the following items when submitting your revised manuscript:* A letter that responds to each point raised by the editor and reviewer(s). You should upload this letter as a separate file labeled 'Response to Reviewers'. This file does not need to include responses to any formatting updates and technical items listed in the 'Journal Requirements' section below.* A marked-up copy of your manuscript that highlights changes made to the original version. You should upload this as a separate file labeled 'Revised Manuscript with Track Changes'.* An unmarked version of your revised paper without tracked changes. You should upload this as a separate file labeled 'Manuscript'. If you would like to make changes to your financial disclosure, competing interests statement, or data availability statement, please make these updates within the submission form at the time of resubmission. Guidelines for resubmitting your figure files are available below the reviewer comments at the end of this letter. We look forward to receiving your revised manuscript. Kind regards, David Fraile Navarro, MD PhDAcademic EditorPLOS Digital HealthDavid Fraile NavarroAcademic EditorPLOS Digital HealthLeo Anthony CeliEditor-in-ChiefPLOS Digital Healthorcid.org/0000-0001-6712-6626**Journal Requirements:**

1. Tables should not be uploaded as individual files. Please remove these files and include the Tables in your manuscript file as editable, cell-based objects. For more information about how to format tables, see our guidelines: [LINK]

https://journals.plos.org/digitalhealth/s/tables

If the reviewer comments include a recommendation to cite specific previously published works, please review and evaluate these publications to determine whether they are relevant and should be cited. There is no requirement to cite these works unless the editor has indicated otherwise. **Additional Editor Comments (if provided):**

This manuscript evaluates automated multi-label crisis detection on 1,057 psychological hotline calls (2022–2023), comparing embedding-based deep learning models (audio/text pretrained representations) with LLM prompt-based prediction, and includes an expert review of generated explanations. However, the following concerns should be addressed before publication:

1.Unclear dataset generation and questionnaire administration. The manuscript defines and computes depression, high-risk status, and risk-factor scores, but does not specify how the 20-item questionnaire was administered in real hotline calls (routine scripted questions vs operator judgment vs post-call coding). This ambiguity directly affects label validity and data quality.

Label-definition mismatch between questionnaire and LLM prompts. “High-risk” is operationalized in the dataset via explicit logical criteria involving suicidal ideation, suicidal plan, and a risk-factor, while the prompt redefines “high-risk” as a broader “immediate and severe crisis” concept with a 7-level ladder later collapsed to binary.

3.Transcript creation and “caller-only” extraction are not reproducible. Transcripts are produced by a third-party ASR system, and the analysis uses caller-only text, but the manuscript does not document speaker separation (diarization/role tagging/heuristics), error rates, or any ASR quality control.

Statistical claims are not supported. The paper reports averages over repeated runs, but variance/CI reporting is inconsistent and “no significant gain” statements for fusion are not backed by clear tests, effect sizes, or confidence intervals. The supplement’s use of independent-samples t-tests across repeated runs is not appropriate for comparing methods on the same fixed test set, and multiple comparisons across many models/labels are not controlled, so p<0.05 claims are not reliable.

5.Case-control sampling limits external validity. The dataset is artificially balanced (roughly half high-risk via matched controls), while real-world hotline prevalence is likely much lower. Precision/recall, thresholds, and operational utility derived from this setting will not transfer without prevalence-aware evaluation.

6.Audio baseline likely handicaps acoustic information. The audio pipeline averages embeddings over 10-second segments, collapsing temporal dynamics that plausibly encode distress escalation and hesitation patterns. Concluding that “acoustic features add negligible value” is not justified without stronger temporal audio modeling or ablations that isolate where acoustic signal is lost.

Missing data in Table 1 (marked with *) must be handled explicitly. Provide denominators for each variable and state a defined missing-data strategy (complete-case vs imputation).

8.Education stratification is not defined. “Low/medium/high” must be explicitly mapped to concrete education categories.

9.Fix typos (including the “.Audio library” and reference), and revise ambiguous sentences in Methods/Results so that data flow, model inputs, and evaluation steps are unambiguous.

**Reviewers' Comments:** Reviewer's Responses to Questions

**Comments to the Author**

1. Does this manuscript meet PLOS Digital Health’s publication criteria? Is the manuscript technically sound, and do the data support the conclusions? The manuscript must describe methodologically and ethically rigorous research with conclusions that are appropriately drawn based on the data presented.

Reviewer #1: Yes

Reviewer #2: Yes

2. Has the statistical analysis been performed appropriately and rigorously?

Reviewer #1: Yes

Reviewer #2: Yes

3. Have the authors made all data underlying the findings in their manuscript fully available (please refer to the Data Availability Statement at the start of the manuscript PDF file)?

Reviewer #1: Yes

Reviewer #2: Yes

4. Is the manuscript presented in an intelligible fashion and written in standard English?

Reviewer #1: Yes

Reviewer #2: Yes

5. Review Comments to the Author

Reviewer #1: This study analyzes psychological crises from the hotline by comparing audio-based solutions and text-based solutions. The study is comprehensive because it covers both audio-based and text-based approaches, and it evaluates both deep learning-based approaches and LLM-based approaches.

Some details of the study need to be further clarified.

Line 35: "Surprisingly, XXX, utilizing few-shot learning, exhibited the highest F1 scores across all multi-label tasks."

Why is this surprising, and how to explain this surprising result?

Line 119: How to define "high-risk calls"? Are they manually annotated?

Line 201: It is interesting to see "multimodal synergy". It is further described in equation (5) on Line 504. However, there is no result for this experiment.

Figure 4: What are rubrics to evaluate label consistency, logical reasoning, information completeness, clinical relevance, and clarity and comprehensibility in the expert evaluation?

Line 472: "combining the reconstruction task of an autoencoder with the classification task" is described in S2. What is the purpose of using an auto-encoder and sharing it with the classification task, instead of using a classification framework only?

The whole manuscript, please change the reference citation from () to [ ].

Reviewer #2: This study presents a rigorous evaluation of pre-trained deep learning models and large language models (LLMs) for multi-label crisis detection in psychological support hotlines. Using 1,057 real-world hotline calls, the authors demonstrate that text-based semantic representations—particularly GPT-derived embeddings—significantly outperform acoustic-only and multimodal approaches in identifying emotional distress and suicide risk, achieving an F1 score of 80.48% for high-risk detection. Notably, the work shows that few-shot LLMs (DeepSeek-R1) can match supervised deep learning performance while requiring minimal labeled data. The integration of expert evaluation of model-generated explanations adds important clinical relevance and interpretability.

Action item:

Validate models on multicenter, culturally diverse hotline datasets to improve generalizability.

Address class imbalance reflective of real-world call distributions.

Incorporate sequential or temporal fusion methods for richer audio modeling.

Develop standardized, objective frameworks for evaluating LLM-generated clinical explanations.

6. PLOS authors have the option to publish the peer review history of their article (what does this mean?). If published, this will include your full peer review and any attached files.

**Do you want your identity to be public for this peer review?** For information about this choice, including consent withdrawal, please see our Privacy Policy.

Reviewer #1: No

Reviewer #2: **Yes:** sandeep chaurasia

  **Figure resubmission:**  While revising your submission, we strongly recommend that you use PLOS’s NAAS tool (https://ngplosjournals.pagemajik.ai/artanalysis) to test your figure files. NAAS can convert your figure files to the TIFF file type and meet basic requirements (such as print size, resolution), or provide you with a report on issues that do not meet our requirements and that NAAS cannot fix. 

After uploading your figures to PLOS’s NAAS tool - https://ngplosjournals.pagemajik.ai/artanalysis, NAAS will process the files provided and display the results in the "Uploaded Files" section of the page as the processing is complete. If the uploaded figures meet our requirements (or NAAS is able to fix the files to meet our requirements), the figure will be marked as "fixed" above. If NAAS is unable to fix the files, a red "failed" label will appear above. When NAAS has confirmed that the figure files meet our requirements, please download the file via the download option, and include these NAAS processed figure files when submitting your revised manuscript. **Reproducibility:** To enhance the reproducibility of your results, we recommend that authors of applicable studies deposit laboratory protocols in protocols.io, where a protocol can be assigned its own identifier (DOI) such that it can be cited independently in the future. Additionally, PLOS ONE offers an option to publish peer-reviewed clinical study protocols. Read more information on sharing protocols at https://plos.org/protocols?utm_medium=editorial-email&utm_source=authorletters&utm_campaign=protocols

---

## [Decision Letter · Decision Letter 1]

7 Apr 2026

Automating Multi-label Crisis Detection in Psychological Support Hotlines with Pre-trained Models

PDIG-D-25-01203R1

Dear Phd Jiang,

We are pleased to inform you that your manuscript 'Automating Multi-label Crisis Detection in Psychological Support Hotlines with Pre-trained Models' has been provisionally accepted for publication in PLOS Digital Health.

Best regards,

David Fraile Navarro, MD PhD

Academic Editor

PLOS Digital Health

**Additional Editor Comments (if provided):**

**Reviewer Comments (if any, and for reference):**

Reviewer's Responses to Questions

**Comments to the Author**

1. If the authors have adequately addressed your comments raised in a previous round of review and you feel that this manuscript is now acceptable for publication, you may indicate that here to bypass the “Comments to the Author” section, enter your conflict of interest statement in the “Confidential to Editor” section, and submit your "Accept" recommendation.

Reviewer #1: All comments have been addressed

2. Does this manuscript meet PLOS Digital Health’s publication criteria? Is the manuscript technically sound, and do the data support the conclusions? The manuscript must describe methodologically and ethically rigorous research with conclusions that are appropriately drawn based on the data presented.

Reviewer #1: Yes

3. Has the statistical analysis been performed appropriately and rigorously?

Reviewer #1: Yes

4. Have the authors made all data underlying the findings in their manuscript fully available (please refer to the Data Availability Statement at the start of the manuscript PDF file)?

Reviewer #1: No

5. Is the manuscript presented in an intelligible fashion and written in standard English?

Reviewer #1: Yes

6. Review Comments to the Author

Reviewer #1: The revision is fine, all comments have been addressed.

7. PLOS authors have the option to publish the peer review history of their article (what does this mean?). If published, this will include your full peer review and any attached files.

**Do you want your identity to be public for this peer review?** For information about this choice, including consent withdrawal, please see our Privacy Policy.

Reviewer #1: No
